# Raman Study of Block Copolymers of Methyl Ethylene Phosphate with Caprolactone and L-lactide

**DOI:** 10.3390/polym14245367

**Published:** 2022-12-08

**Authors:** Sergei O. Liubimovskii, Vasiliy S. Novikov, Andrey V. Shlyakhtin, Vladimir V. Kuzmin, Maria M. Godyaeva, Sergey V. Gudkov, Elena A. Sagitova, Leila Yu. Ustynyuk, Goulnara Yu. Nikolaeva

**Affiliations:** 1Prokhorov General Physics Institute of the Russian Academy of Sciences, Vavilov Str. 38, 119991 Moscow, Russia; 2Department of Chemistry, M.V. Lomonosov Moscow State University, Leninskie Gory 1-3, 119991 Moscow, Russia; 3A.V. Topchiev Institute of Petrochemical Synthesis of the Russian Academy of Sciences, Leninsky Avenue 29, 119991 Moscow, Russia; 4Soil Science Faculty, M.V. Lomonosov Moscow State University, Leninskie Gory 1-12, 119991 Moscow, Russia; 5Federal Scientific Agronomic and Engineering Center VIM, 1st Institutsky Proezd 5, 109428 Moscow, Russia; 6Federal Research Center of Problems of Chemical Physics and Medicinal Chemistry of the Russian Academy of Sciences, Academician Semenov Avenue 1, 142432 Chernogolovka, Russia

**Keywords:** Raman spectroscopy, methyl ethylene phosphate, caprolactone, L-lactide, block copolymers

## Abstract

We present an in-depth analysis of Raman spectra of novel block copolymers of methyl ethylene phosphate (MeOEP) with caprolactone (CL) and L-lactide (LA), recorded with the excitation wavelengths of 532 and 785 nm. The experimental peak positions, relative intensities and profiles of the poly(methyl ethylene phosphate) (PMeOEP), polycaprolactone (PCL) and poly(L-lactide) (PLA) bands in the spectra of the copolymers and in the spectra of the PMeOEP, PCL and PLA homopolymers turn out to be very similar. This clearly indicates the similarity between the conformational and phase compositions of PMeOEP, PCL and PLA parts in molecules of the copolymers and in the PMeOEP, PCL and PLA homopolymers. Experimental ratios of the peak intensities of PMeOEP bands at 737 and 2963 cm^−1^ and the PCL bands at 1109, 1724 and 2918 cm^−1^ can be used for the estimation of the PCL—b—PMeOEP copolymers chemical composition. Even though only one sample of the PMeOEP—b—PLA copolymers was experimentally studied in this work, we assume that the ratios of the peak intensities of PLA bands at 402, 874 and 1768 cm^−1^ and the PMeOEP band at 737 cm^−1^ can be used to characterize the copolymer chemical composition.

## 1. Introduction

Besides production of various ecologically friendly consumer goods, biodegradable and biocompatible polymers have great potential in the state-of-the-art biomedical and pharmaceutical applications, such as creating of the nanocarriers for targeted drug delivery with controlled release of active substances, implants as well as functional coatings on implants, scaffolds for tissue engineering, suture materials, etc. [1,2]. Such polymer items as well as products of their degradation should be non-toxic, degrade in vivo in human organism in particular time and with particular rate (with particular profile of degradation), and the products of the degradation should be eliminated from the body without any harmful impact for organism. Water-soluble synthetic biocompatible polymers have a great significance in the described biomedical applications [3,4,5,6].

Polyethylene glycol (PEG)—the product of ethylene oxide polymerization—perhaps is the most widely used hydrophilic polymer in biomedical field. PEGylation is the commonly used method for reducing cell adhesion and stealth effect creation. PEGylation allows to increase the circulation time of nanoparticles in blood and so reduces the drug dose [7,8]. However, the non-biodegradability and immunogenicity of PEG have triggered a search for alternatives [5,9].

Synthetic polyphosphoesters have been studied since the 1950s, but they have been systematically investigated for biomedical application over the past 30 years. Polyethylene phosphates such as PEG possess a stealth effect, but do not accumulate in the body due to biodegradation. Thanks to controlled synthesis by ring-opening polymerization (ROP), structural variability, biocompatibility and biodegradability, polyphosphates possess a wide range of applications such as construction of medical items with adjusting biocompatibility, hydrophobicity and biodegradation time [9,10,11].

Amphiphilic block copolymers of methyl ethylene phosphate (MeOEP) with caprolactone (CL) and L-lactide (LA) are very promising for biomedical applications due to the possibility of varying their physical and chemical properties, including the degradation profile, in wide range by alteration of the conditions of synthesis or post-treatment. The presence of hydrophilic poly(methyl ethylene phosphate) (PMeOEP) and hydrophobic polycaprolactone (PCL) or poly(L-lactide) (PLA) parts makes possible the self-organization of the copolymer molecules in nanostructures of various architectures [1,2,3]. Since the structure of a polymer material determines its properties, it is very important to have an informative and fast method of evaluating the most important structural characteristics of polymer items without any preparation of samples for analysis, because such preparation can affect the sample structure.

Raman spectroscopy is a highly informative and non-destructive technique for analyzing all levels of polymer structure, including evaluation of the degree of crystallinity, tacticity, conformational composition and molecular orientation. This method is very effective for description of both crystalline and non-crystalline regions of polymer material. Raman spectroscopy can also provide *in situ* analysis of polymerization and degradation processes and fast structural mapping of sample surface. This technique does not require any sample preparation before the analysis.

Among the three polymers (PLA, PCL, and PMeOEP), PLA is the most studied by Raman spectroscopy. It was shown that the Raman spectra of PLA depend on the degree of crystallinity, tacticity and conformational composition of molecules [12,13,14], on molecular weight [15,16], and on the orientational order of molecules [14,17,18].

There is a limited amount of research reporting Raman spectra of copolymers of LA. In particular, Kister et al. published Raman spectra of poly(L-lactide-co-glycolide), poly(D,L-lactide-co-glycolide) [19] and poly(ε-caprolactone-co-D,L-lactide) [20]. Several studies have presented results of Raman spectroscopy applied for monitoring the degradation process of PLA and PLA-based materials [20,21,22]. Quantum chemical calculations were applied to study the structure and Raman spectra of PLA oligomers [16] and various crystalline modifications of PLA [23].

Few papers on Raman studies of PCL are found in literature. In particular, Kotula et al. [24] demonstrated the dependence of PCL Raman spectra on the degree of crystallinity and conformational order. We could not find any other works on systematic studies of PMeOEP Raman spectra.

Despite the many published papers on Raman studies of PLA, up to now there has been no full assignment of Raman bands in the spectra of various modifications of PLA [25] as well as Raman methods for quantitative PLA structure analysis and PLA-based materials. As it was mentioned before, the Raman spectra of PCL and PMeOEP are poorly studied. Moreover, Raman spectroscopy has still not been used in studying the block copolymers of MeOEP with CL or LA. The aim of this work is to examine the possibilities of Raman spectroscopy for quantitative description of the structure of these copolymers.

## 2. Materials and Methods

### 2.1. General Experimental Remarks

Toluene, diethyl ether and tetrahydrofuran (THF) were refluxed with Na/benzophenone/dibenzo-18-crown-6 and distilled prior to use. Dichloromethane was refluxed over CaH_2_ prior to use.

Acetic acid (Acros, ≥99.9%) was used as purchased. Poly(ethylene glycol) methyl ether (average M_n_ 550 Da, Sigma-Aldrich, St. Louis, MO, USA, 97%) was dried over 4 Å molecular sieves overnight prior to use. Poly(ethylene glycol) methyl ethers (average M_n_ 2000 Da, 5000 Da, Sigma-Aldrich, St. Louis, MO, USA, 97%) were used as purchased. (*S*,*S*)-3,6-Dimethyl-1,4-dioxane-2,5-dione (L-LA) (Sigma-Aldrich, St. Louis, MO, USA, 99%) was purified by recrystallization from dry toluene and subsequent sublimation and stored over P_2_O_5_. Caprolactone (CL) (Sigma-Aldrich, St. Louis, MO, USA, 97%) was distilled over CaH_2_ prior to use.

[(BHT)Mg(OBn)(THF)]_2_ (BHT = 2,6-di-tret-butyl-4-methylphenoxy, Bn = PhCH_2_—), Mg1 [26], [(BHT)Mg(n-Bu)(THF)_2_], Mg2 [27], methyl ethylene phosphate (MeOEP) [28] were prepared according to previously described methods.

CDCl_3_ (Cambridge Isotope Laboratories, Inc., Tewksbury, MA, USA, D 99.8%) was used as purchased. The ^1^H NMR spectra were recorded on a Bruker AVANCE 400 spectrometer (400 MHz) at 20 °C.

### 2.2. Polymers Synthesis

Polymerization experiments were conducted under an argon atmosphere. Polyesters were obtained by ring-opening polymerization of cyclic esters and cyclic phosphates (Figure 1). ROP is a convenient method for the synthesis of polyesters, which allows to control the molecular weight of polyesters, as well as to obtain the block copolymers by sequentially adding comonomers [29,30,31]. A phenolate–alcohol–magnesium complex Mg1 was used as a catalyst for ROP (Figure 2a), which also acts as an initiator of ROP [32,33,34,35,36,37,38]. To synthesize methylated PEG (mPEG)-based polymers, a phenolate-mPEG-magnesium complex was obtained *in situ* from monomethyl ether of polyethylene glycol mPEG-OH and an organometallic phenolate–magnesium complex Mg2 (Figure 2b).

Table 1 presents a chemical composition of all polymers under study, determined by ^1^H NMR spectroscopy. As for PCL—b—PMeOEP and PMeOEP—b—PLA copolymers, the PMeOEP percent in the designation of each sample (the first column in Table 1) does not include the contribution of mPEGs or BnOH.

For more details on the synthesis of polymers under study, see Appendix A. Synthesis in the Appendix A.

### 2.3. Raman Measurements

The Raman spectra were acquired at room temperature using a confocal Raman microscope Senterra II (Bruker Optics, Billerica, MA, USA) with the excitation wavelengths of 532 and 785 nm. Spectra were recorded at 180°-scattering with the spectral resolution of 4 cm^−1^. Laser power at sample surface was 25 mW for the excitation wavelength of 532 nm and 100 mW—for 785 nm. For all the measurements, we used the 20× objective with NA = 0.4. The laser spot at the sample surface was about 10 μm for the excitation wavelength of 532 nm and about 12 μm for the excitation wavelength of 785 nm.

We measured Raman spectra at several points at the surface of each sample in order to be sure that the structure of the samples is uniform over the sample surface. In addition, we recorded Raman spectra of each sample several times to ensure that the structure of the samples is not changing during the recording of Raman spectra due to heating.

For a number of samples, we registered a strong fluorescence background, which mainly observed the excitation wavelength of 532 nm and rarely observed the excitation wavelength of 785 nm. Probably, this background can be explained by presence of the impurities, remaining after synthesis. In all the spectra with the strong fluorescence background we also observed additional broad features—«ripples» [39]. The peak positions, relative intensities and profiles of these «ripples» are repeated in all the spectra with a high fluorescence background, while these additional features are unnoticeable in all the spectra with a low fluorescence background. These «ripples» were explained by the presence of an edge-filter in Raman setups [39], and there are no possibilities to avoid their appearance in the spectra of highly fluorescing samples for the Raman setup used. These artifacts can influence the analysis of weak Raman bands and, thus, careful examination of weak Raman bands is required in the case of presence of strong fluorescence background, when the «ripples» are clearly pronounced.

### 2.4. Quantum Chemical Calculations

To assign the Raman bands of PLA, PCL and PMeOEP, we performed the quantum chemical calculations of the optimized geometries and Raman spectra of oligomers of these polymers, using as the models oligomers consisting of from 4 to 9 monomeric units (for details, see Appendix A).

For the calculations, we used the DFT method, the software package PRIRODA [40], the harmonic oscillator approximation, the OLYP functional [41] and 4z contracted set of Gaussian basis functions (for details, see Appendix A), implemented in the PRIRODA program. In our previous work [42], this combination of the functional and the basis set was found to provide good agreement of the results of calculations with the experimental Raman frequencies and intensities.

Table 2 presents the assignment of the Raman bands of PLA, PCL and PMeOEP, obtained from the results of the DFT analysis with the help of the Chemcraft Lite program [43].

## 3. Results and Discussion

### 3.1. Raman Spectra of PCL Samples

Figure 3 and Figure 4 demonstrate the Raman spectra of six PCL samples, recorded with the excitation wavelengths of 532 nm (Figure 3) and 785 nm (Figure 4). Here and further, the spectra are presented in two spectral regions: 200–1850 and 2600–3400 cm^−1^ after subtraction of the fluorescence background and normalization to the maximal intensity in each shown spectral range. The additional spectral features—«ripples» (see Section 2.3. Raman measurements) were not removed during processing of the spectra in order to exclude an influence of the complex background subtraction on weak bands of the homopolymers and copolymers. There are no noticeable Raman bands in the region 1850–2600 cm^−1^ of the spectra of all the samples under study, thus, this spectral range is not considered in this work.

The synthesis of the PCL samples differs in the type, the contents of catalyst and the initiator (see Section 2.2. Polymers synthesis). It can be seen from Figure 3 and Figure 4 that the peak positions, relative intensities and profiles of all the PCL bands are the same in the spectra of these six samples. This means that the conformational and phase compositions of PCL molecules are very similar for all these PCL samples. However, the spectra of PCL1, PCL2 and PCL3 contain evident additional bands, which have the same peak positions for both excitation wavelengths. We assigned these additional bands to the Raman spectra of the initiators: mPEG5000 in the case of PCL1 and BnOH in the case of PCL2 and PCL3. These Raman bands of the initiators in the PCL spectra will be discussed in detail in Section 3.4. Raman bands of the initiators in the spectra of PCL samples.

### 3.2. Raman Spectra of PMeOEP Samples

Figure 5 and Figure 6 show Raman spectra of two PMeOEP samples, which contain different amounts of BnOH (see Section 2.2. Polymers synthesis), for the two excitation wavelengths: 532 nm (Figure 5) and 785 nm (Figure 6). The spectra of these two samples are identical for the excitation wavelength of 785 nm. However, for the excitation wavelength of 532 nm the spectrum of PMeOEP2 contains evident additional bands, which were related to the «ripples», caused by the high intensity of the fluorescence background and the presence of an edge-filter in the Raman setup used [39]. These additional bands do not correspond to the Raman spectrum of BnOH, which will be discussed in the next Section 3.3. Raman spectra of neat initiators: mPEGs and BnOH. The Raman spectrum of PMeOEP1, recorded with the excitation wavelength of 532 nm, also contains the «ripples», but their intensities are much smaller than in the case of the spectrum of PMeOEP2.

### 3.3. Raman Spectra of Neat Initiators: mPEGs and BnOH

Figure 7 and Figure 8 demonstrate the Raman spectra of the initiators: BnOH and mPEGs, which were used for the synthesis in this study. The spectra were recorded using two excitation wavelengths: 532 nm (Figure 7) and 785 nm (Figure 8). It can be seen that the spectra of two solid mPEGs (mPEG2000 and mPEG5000) are evidently different. Furthermore, the spectra of the solid mPEGs differ noticeably from the spectrum of liquid mPEG (mPEG550), but these differences are due to the different aggregate states rather than to the different molecular weights or change in the conformational composition of molecules. In particular, it was recently shown [44] that at room temperature, molecules of both liquid and solid PEGs with various molecular weights adopt predominantly the conformation of helix 7_2_.

It is worth mentioning that the Raman spectra of solid non-methylated PEGs do not depend on the molecular weight [44,45]. Probably, differences in the Raman spectra of the solid methylated PEGs—mPEG2000 and mPEG5000—can be explained by change in the content of the CH_3_ groups relative to the PEG monomer unit content.

### 3.4. Raman Bands of the Initiators in the Spectra of PCL Samples

Figure 9 demonstrates the Raman spectra of PCL1, PCL6 and mPEG5000, recorded with the excitation wavelengths of 532 nm (a) and 785 nm (b) in the region 200–1850 cm^−1^. PCL1 contains the maximal amount of an initiator among all the PCL samples under the study: mPEG5000 with the mole content about 1.6%, while PCL6 contains the minimal amount of an initiator: mPEG2000 with the mole content about 0.2% (see Section 2.2. Polymers synthesis). Despite the low content of mPEG5000 in PCL1, the mPEG5000 bands are clearly distinguishable in the PCL1 spectrum at both excitation wavelengths. The arrows in Figure 9 mark these bands.

Figure 10 presents the Raman spectra of PCL2, PCL3, PCL6 and BnOH, recorded with the excitation wavelengths of 532 nm (a) and 785 nm (b) in the region 200–1850 cm^−1^. The arrows show the BnOH band at 1005 cm^−1^ (Figure 7 and Figure 8), which is observed in the PCL2 and PCL3 spectra. The mole content of BnOH in PCL2 and PCL3 is less than 1%, but the BnOH band at 1005 cm^−1^ is well resolved in the spectra of these PCL samples. This observation can be explained by the very high intensity of the BnOH band at 1005 cm^−1^ in the spectrum of neat BnOH (Figure 7 and Figure 8).

Thus, we observed Raman bands of the initiators (mPEGs and BnOH) in the Raman spectra of the PCL samples even at the initiator mole contents, which are less than 2%. This value is quite a good detection limit for the non-resonance Raman spectroscopy, applied in this study.

### 3.5. Raman Spectra of the PCL—b—PMeOEP Copolymers

Figure 11 and Figure 12 show the Raman spectra of the PCL—b—PMeOEP copolymers, recorded with the excitation wavelengths of 532 nm (Figure 11) and 785 nm (Figure 12). For comparison, we also present the Raman spectra of PMeOEP1 and PCL6. The PCL6 sample was chosen as the reference sample, because, as it was mentioned above, it contains the minimal amount of an initiator among all the PCL samples under the study. The content of the initiator in PMeOEP1 is higher than in PMeOEP2, but intensities of the «ripples» in the PMeOEP1 spectrum are much less compared to the PMeOEP2 spectrum (see the Section 3.2. Raman spectra of PMeOEP samples). Thus, the PMeOEP1 spectrum was chosen as the reference spectrum for all the copolymers of MeOEP studied in this work. The arrows in Figure 11 and Figure 12 show the PMeOEP bands, observed in the spectra of the copolymers.

The peak positions, relative intensities and profiles of the PMeOEP and PCL bands are nearly the same in the spectra of the copolymers (Figure 11 and Figure 12) and in the spectra of the PMeOEP and PCL samples (Figure 3, Figure 4, Figure 5 and Figure 6). It means that the conformational and phase compositions of the PMeOEP and PCL parts in the copolymer molecules do not differ significantly from that of the PMeOEP and PCL samples.

It can be seen from Figure 11 and Figure 12 that the intensities of the PMeOEP bands in the spectra of the copolymers monotonically decrease with the decrease in the PMeOEP content. Table 3 and Table 4 summarize the ratios of the peak intensities of selected PMeOEP and PCL bands in the spectra of the copolymers for the excitation wavelengths of 532 nm (Table 3) and 785 nm (Table 4). For this analysis, we chose the following bands: the PMeOEP bands at 737 and 2963 cm^−1^ and the PCL bands at 1109, 1724 and 2918 cm^−1^. Error of measurement of the peak intensities ratios was estimated from 5 to 15% depending on the intensities of the analyzed bands in the copolymer spectra and the intensity of the fluorescence background.

It is important to note that for our analysis of the peak intensities ratios, we tried to choose the strongest bands of each polymer, which do not overlap with the bands of the second polymer. However, with the only exception of the PCL band at 1724 cm^−1^, we observed non-zero Raman intensity in the PMeOEP spectrum at the wavenumbers of all the intensive PCL bands, and *vice versa*. Thus, we selected the bands of each of the two polymers in such a manner such that the intensities at the wavenumbers of these selected bands in the spectrum of the second polymer would be minimal. This means that for such bands, the intensity ratios differ from zero even in the case, when the content of the monomer, corresponding to the numerator in the intensity ratio, is zero. Thus, for precise determination of the dependences of the intensity ratios on the relative contents of PMeOEP and PCL in the copolymers, it is necessary to also calculate the intensity ratios in the spectra of each of these two polymers using the wavenumbers of the selected bands.

It is worth noting that we did not perform deconvolution analysis of the copolymers’ Raman spectra. Firstly, this procedure makes calculation of the intensity ratios much more complicated. Secondly, in the spectral regions of the most intense bands of the copolymers, we observed strong overlapping of numerous bands, and the deconvolution analysis was not reliable.

It can be seen from Table 3 and Table 4 that the peak intensities ratios of the PMeOEP and PCL bands in the spectra of the copolymers depend strongly on the relative contents of PMeOEP and PCL and, thus, can be used for estimation of the chemical composition of the PCL—b—PMeOEP copolymers.

### 3.6. Raman Spectra of PLA Samples

Figure 13 and Figure 14 show the Raman spectra of six PLA samples, recorded with the excitation wavelengths of 532 nm (Figure 13) and 785 nm (Figure 14). The synthesis of the PLA samples differs in the type and contents of catalyst and initiator (see Section 2.2. Polymers synthesis). However, the Raman spectra of all the PLA samples turned out to be very similar. This is evidence of the similarity in the phase and conformational compositions of these samples. The only exception was the presence of additional weak band at 1005 cm^−1^ in the PLA1, PLA2 and PLA3 spectra. We assigned this band to a very strong band in the spectrum of neat BnOH (Figure 7 and Figure 8). This band is discussed in detail in the next section.

### 3.7. Raman Bands of the Initiator in the Spectra of PLA Samples

Figure 15 demonstrates Raman spectra of PLA1, PLA2, PLA3, PLA5 and BnOH, recorded with the excitation wavelengths of 532 nm (a) and 785 nm (b) in the region 200–1850 cm^−1^. The arrows show the BnOH band at 1005 cm^−1^ (Figure 7 and Figure 8), observed in the PLA1, PLA2 and PLA3 spectra. The PLA5 spectrum was selected as the reference spectrum because the PLA5 sample contains a minimal amount of the initiators among all the PLA samples under study. The mole content of BnOH in PLA1, PLA2 and PLA3 is less than 2%, but the BnOH band at 1005 cm^−1^ is well resolved in the spectra of these samples.

Thus, as in the case of PCL (the Section 3.4. Raman bands of the initiators in the spectra of PCL samples), we observed Raman bands of the initiator (BnOH) in the Raman spectra of PLA samples even at the initiator mole contents, which are less than 2%.

### 3.8. Raman Spectra of the PMeOEP—b—PLA Copolymer

Figure 16 and Figure 17 show the Raman spectra of the PMeOEP—b—PLA copolymer as well as the reference spectra of PMeOEP1 and PLA5, recorded with the excitation wavelength of 532 nm (Figure 16) and 785 nm (Figure 17). The arrows show the PMeOEP bands, observed in the copolymer spectrum.

The spectral characteristics (peak positions, relative intensities and profiles) of the PMeOEP and PLA bands are nearly the same in the copolymer spectrum and in the spectra of the PMeOEP and PLA samples (Figure 5, Figure 6, Figure 13 and Figure 14). It means that the conformational and phase compositions of the PMeOEP and PLA molecules are close for the copolymer and the PMeOEP and PLA samples. 

Although only one sample of the PMeOEP—b—PLA copolymer is available for the analysis in this work, we can propose the PMeOEP and PLA bands, which can be used to estimate the relative contents of PMeOEP and PLA. The strongest bands of PLA, which do not significantly overlap with the intensive PMeOEP bands, are observed at 402, 874 and 1768 cm^−1^. We have found only one strong PMeOEP band at 737 cm^−1^, which does not overlap with the intensive PLA bands. Thus, in the case that the conformational and phase compositions of copolymer molecules do not change noticeably with variation in the relative contents of PMeOEP and PLA, these four bands can be used to estimate the copolymer chemical composition.

## 4. Conclusions

In this work, we analyzed experimental Raman spectra of novel and practically important materials: block copolymers of methyl ethylene phosphate (MeOEP) with caprolactone (CL) and L-lactide (LA). Analysis of the peak positions, relative intensities and profiles of the poly(methyl ethylene phosphate) (PMeOEP), polycaprolactone (PCL) and poly(L-lactide) (PLA) bands showed that the conformational and phase compositions of PMeOEP, PCL and PLA parts in the copolymers molecules do not differ significantly from those of the PMeOEP, PCL and PLA homopolymers samples.

The ratios of intensities of the PMeOEP and PCL bands in the spectra of the PCL—b—PMeOEP copolymers can be used for estimation of the chemical composition of the copolymers. If the conformational and phase compositions of PMeOEP, PCL and PLA parts of the copolymer molecules are close to those of the homopolymers, as in our case, we suggest using the following bands — the PMeOEP bands at 737 and 2963 cm^−1^ and the PCL bands at 1109, 1724 and 2918 cm^−1^. However, for more general consideration, further studies should involve accounting for the influence of possible changes of the conformational composition and the degree of crystallinity of the copolymers with the change in the comonomer contents.

The PLA bands at 402, 874 and 1768 cm^−1^ and the PMeOEP band at 737 cm^−1^ seem to be attractive for evaluation of chemical composition of the PMeOEP—b—PLA copolymers.

The research results can be used for fast and non-destructive estimation of the chemical composition of the PCL—b—PMeOEP and PMeOEP—b—PLA copolymers. The developed approach can be especially in demand during the online synthesis monitoring, or when the fast surface mapping of a copolymer sample is required.

The data on the conformational and phase copolymer composition are useful for prediction and explanation of their physical and chemical properties.

## Figures and Tables

**Figure 1 polymers-14-05367-f001:**
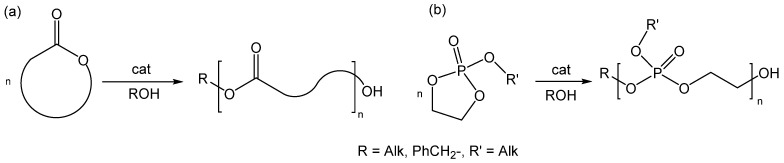
ROP of lactones (**a**) and five-membered ring phosphates (**b**).

**Figure 2 polymers-14-05367-f002:**
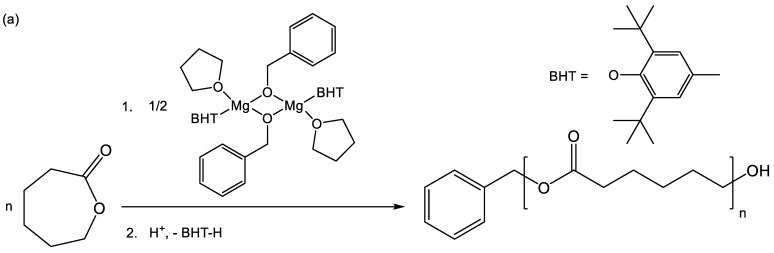
CL polymerization catalyzed by phenolate–benzyloxy–magnesium complex (**a**). Synthesis of mPEG-based block-copolymer catalyzed by phenolate-mPEG-magnesium made in situ (**b**).

**Figure 3 polymers-14-05367-f003:**
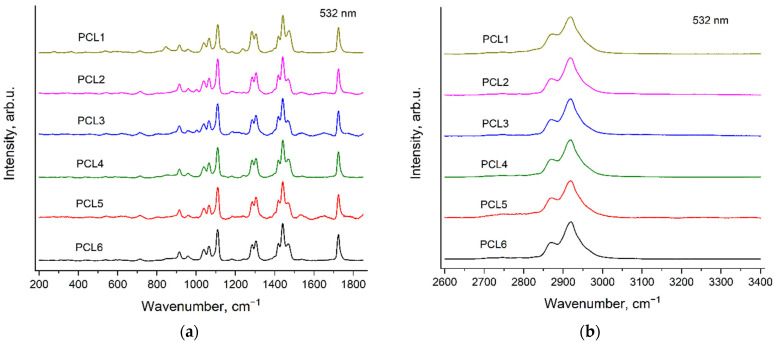
Raman spectra of six PCL samples (see Section 2.2. Polymers synthesis), recorded with the excitation wavelength of 532 nm in the regions 200–1850 cm^−1^ (**a**) and 2600–3400 cm^−1^ (**b**).

**Figure 4 polymers-14-05367-f004:**
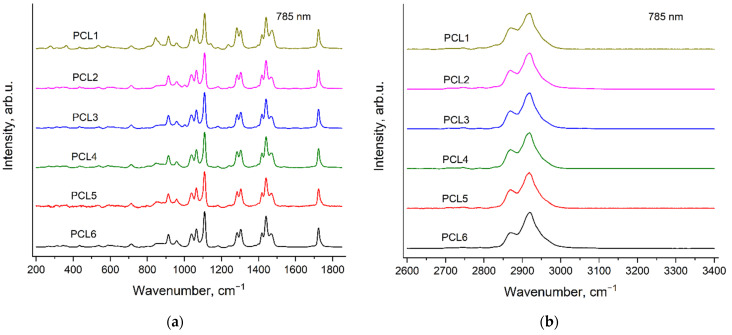
Raman spectra of six PCL samples (see Section 2.2. Polymers synthesis), recorded with the excitation wavelength of 785 nm in the regions 200–1850 cm^−1^ (**a**) and 2600–3400 cm^−1^ (**b**).

**Figure 5 polymers-14-05367-f005:**
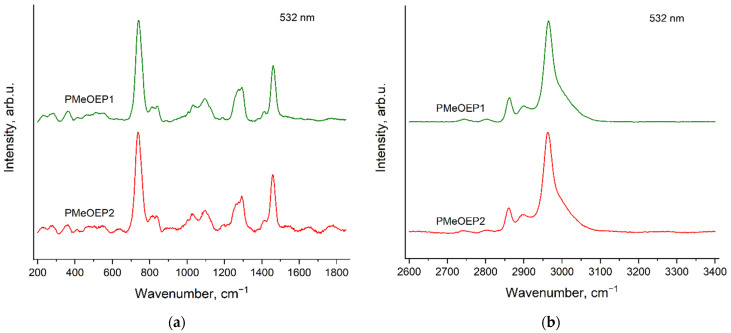
Raman spectra of two PMeOEP samples (see Section 2.2. Polymers synthesis), recorded with the excitation wavelength of 532 nm in the regions 200–1850 cm^−1^ (**a**) and 2600–3400 cm^−1^ (**b**).

**Figure 6 polymers-14-05367-f006:**
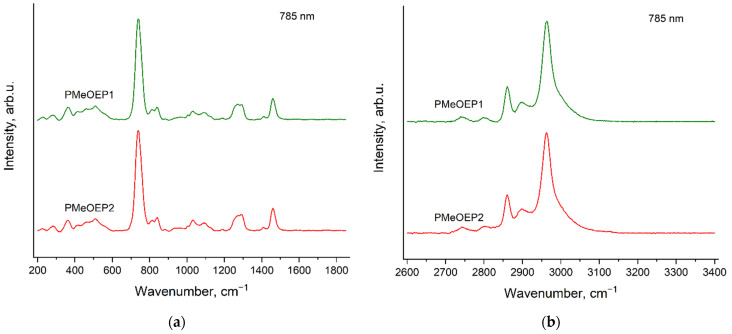
Raman spectra of two PMeOEP samples (see Section 2.2. Polymers synthesis), recorded with the excitation wavelength of 785 nm in the regions 200–1850 cm^−1^ (**a**) and 2600–3400 cm^−1^ (**b**).

**Figure 7 polymers-14-05367-f007:**
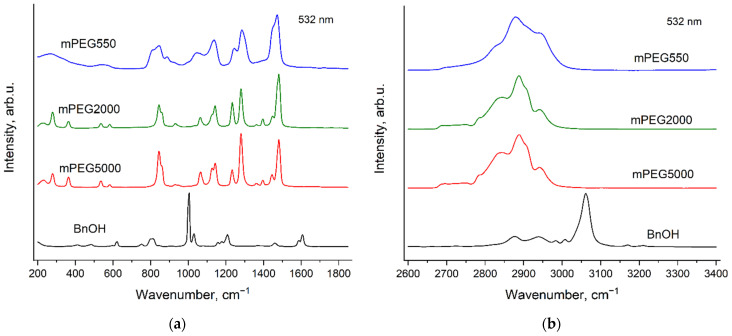
Raman spectra of BnOH and mPEGs with various molecular weights, recorded with the excitation wavelength of 532 nm in the regions 200–1850 cm^−1^ (**a**) and 2600–3400 cm^−1^ (**b**).

**Figure 8 polymers-14-05367-f008:**
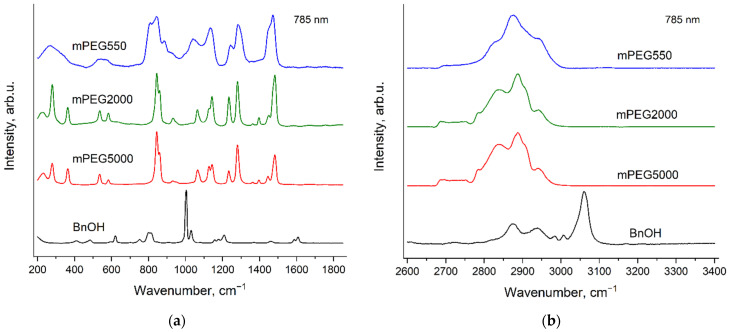
Raman spectra of BnOH and mPEGs with various molecular weights, recorded with the excitation wavelength of 785 nm in the regions 200–1850 cm^−1^ (**a**) and 2600–3400 cm^−1^ (**b**).

**Figure 9 polymers-14-05367-f009:**
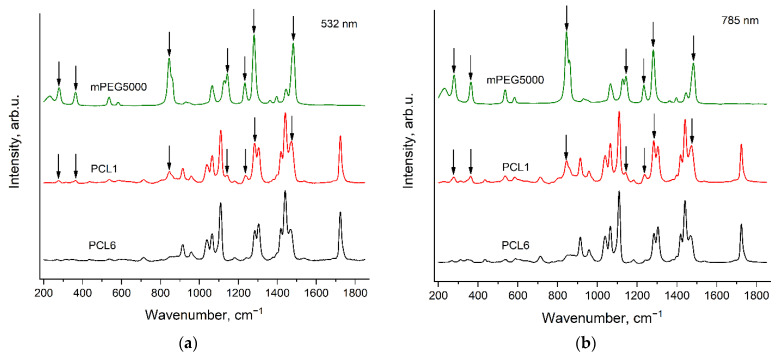
Raman spectra of PCL1, PCL6 (see Section 2.2. Polymers synthesis) and mPEG5000, recorded with the excitation wavelengths of 532 nm (**a**) and 785 nm (**b**) in the region 200–1850 cm^−1^. The arrows show the mPEG5000 bands, observed in the PCL1 spectrum.

**Figure 10 polymers-14-05367-f010:**
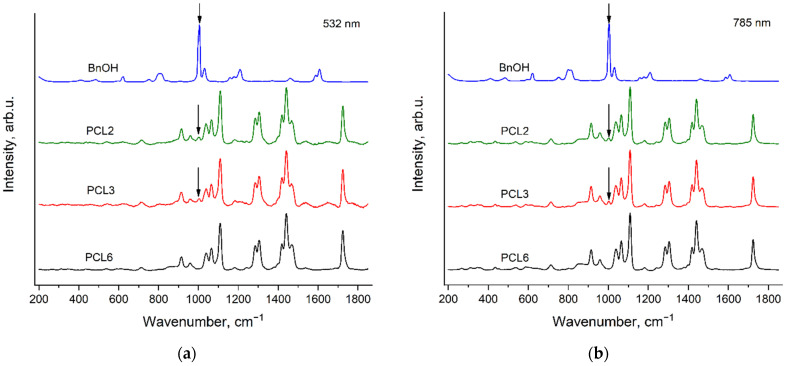
Raman spectra of PCL2, PCL3, PCL6 (see Section 2.2. Polymers synthesis) and BnOH, recorded with the excitation wavelengths of 532 nm (**a**) and 785 nm (**b**) in the region 200–1850 cm^−1^. The arrows show the BnOH band, observed in the PCL2 and PCL3 spectra.

**Figure 11 polymers-14-05367-f011:**
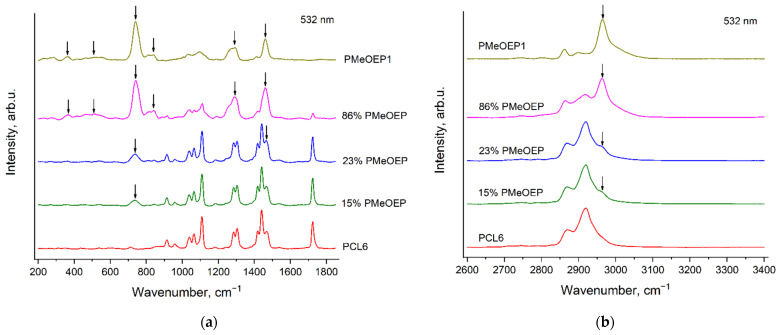
Raman spectra of PMeOEP1, PCL6 and PCL—b—PMeOEP copolymers (see Section 2.2. Polymers synthesis), recorded with the excitation wavelength of 532 nm in the regions 200–1850 cm^−1^ (**a**) and 2600–3400 cm^−1^ (**b**). The arrows show the PMeOEP bands, observed in the spectra of the copolymers.

**Figure 12 polymers-14-05367-f012:**
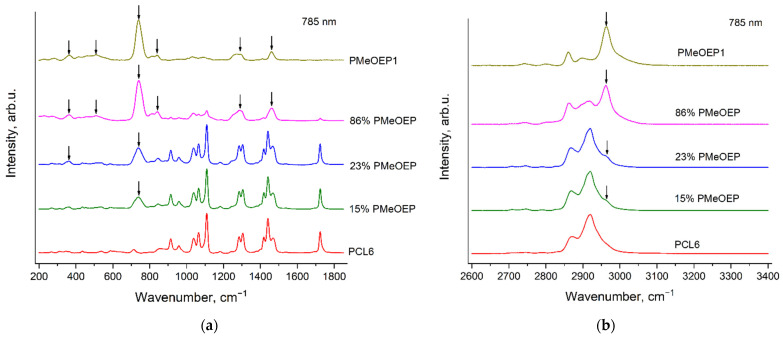
Raman spectra of PMeOEP1, PCL6 and PCL—b—PMeOEP copolymers (see Section 2.2. Polymers synthesis), recorded with the excitation wavelength of 785 nm in the regions 200–1850 cm^−1^ (**a**) and 2600–3400 cm^−1^ (**b**). The arrows show the PMeOEP bands, observed in the spectra of the copolymers.

**Figure 13 polymers-14-05367-f013:**
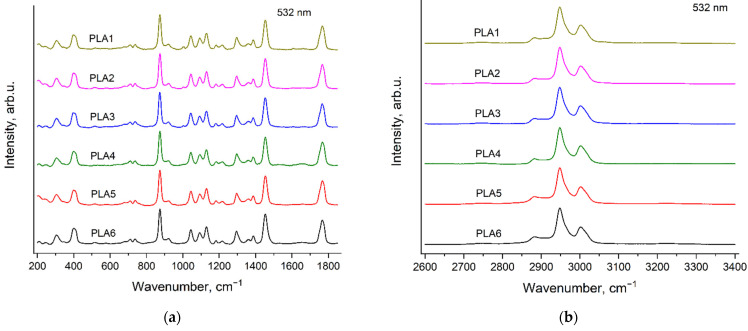
Raman spectra of six PLA samples (see Section 2.2. Polymers synthesis), recorded with the excitation wavelength of 532 nm in the regions 200–1850 cm^−1^ (**a**) and 2600–3400 cm^−1^ (**b**).

**Figure 14 polymers-14-05367-f014:**
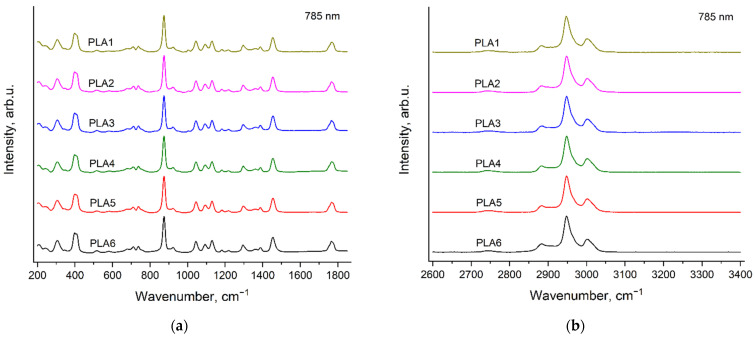
Raman spectra of six PLA samples (see Section 2.2. Polymers synthesis), recorded with the excitation wavelength of 785 nm in the regions 200–1850 cm^−1^ (**a**) and 2600–3400 cm^−1^ (**b**).

**Figure 15 polymers-14-05367-f015:**
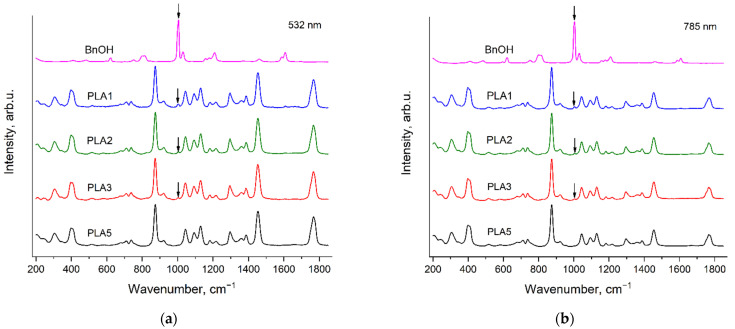
Raman spectra of PLA1, PLA2, PLA3, PLA5 (see Section 2.2. Polymers synthesis) and BnOH, recorded with the excitation wavelengths of 532 nm (**a**) and 785 nm (**b**) in the region 200–1850 cm^−1^. The arrows show the BnOH band, observed in the PLA1, PLA2 and PLA3 spectra.

**Figure 16 polymers-14-05367-f016:**
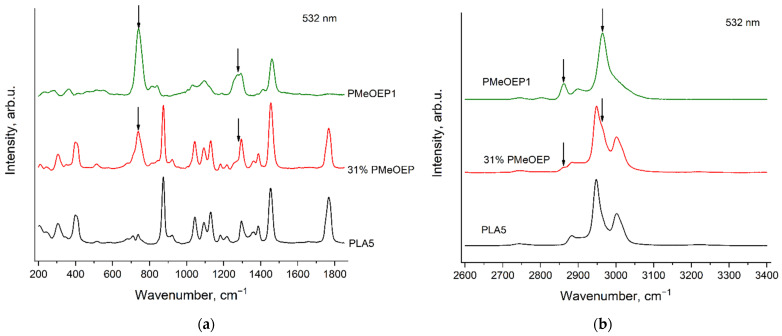
Raman spectra of PMeOEP1, PLA5 and PMeOEP—b—PLA copolymer (see Section 2.2. Polymers synthesis), recorded with the excitation wavelength of 532 nm in the regions 200–1850 cm^−1^ (**a**) and 2600–3400 cm^−1^ (**b**). The arrows show the PMeOEP bands, observed in the copolymer spectrum.

**Figure 17 polymers-14-05367-f017:**
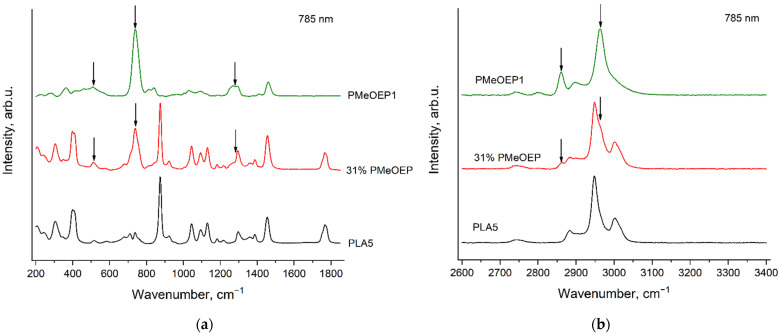
Raman spectra of PMeOEP1, PLA5 and PMeOEP—b—PLA copolymer (see Section 2.2. Polymers synthesis), recorded with the excitation wavelength of 785 nm in the regions 200–1850 cm^−1^ (**a**) and 2600–3400 cm^−1^ (**b**). The arrows show the PMeOEP bands, observed in the copolymer spectrum.

**Table 1 polymers-14-05367-t001:** Chemical composition of the polymers under study, determined by ^1^H NMR spectroscopy.

Samples	Composition (NMR Spectroscopy Data, Relative Content of the Units)						
	mPEG550-	mPEG2000-	mPEG5000-	BnO-	CL	MeOEP	LA
**PCL**							
**PCL1**	—	—	1	—	60	—	—
**PCL2**	—	—	—	1	117	—	—
**PCL3**	—	—	—	1	120	—	—
**PCL4**	—	1	—	—	157	—	—
**PCL5**	—	1	—	—	354	—	—
**PCL6**	—	1	—	—	506	—	—
**PMeOEP**							
**PMeOEP1**	—	—	—	1	—	92	—
**PMeOEP2**	—	—	—	1	—	131	—
**PCL—b—PMeOEP copolymers**							
**5% PMeOEP**	—	—	—	1	264	14	—
**6% PMeOEP**	—	—	—	1	295	19	—
**15% PMeOEP**	—	—	—	1	327	56	—
**16% PMeOEP**	—	—	—	1	342	63	—
**23% PMeOEP**	1	—	—	—	101	31	—
**86% PMeOEP**	1	—	—	—	5	30	—
**PLA**							
**PLA1**	—	—	—	1	—	—	64
**PLA2**	—	—	—	1	—	—	118
**PLA3**	—	—	—	1	—	—	121
**PLA4**	—	—	—	1	—	—	500
**PLA5**	—	—	—	1	—	—	622
**PLA6**	—	—	1	—	—	—	427
**PMeOEP—b** **—PLA copolymer**							
**31% PMeOEP**	—	—	—	1	—	50	112

**Table 2 polymers-14-05367-t002:** Assignment of the Raman bands of PLA, PCL and PMeOEP, obtained from the results of the DFT analysis.

Wavenumber, cm^−1^	Assignment
**PMeOEP**	
**737**	Symmetric stretching vibrations of PO_4_ groups
**1459**	Scissoring vibrations of CH_2_ and CH_3_ groups
**2963**	Symmetric stretching vibrations of CH_2_ and CH_3_ groups
**PCL**	
**1109**	Stretching vibrations of C-C bonds in the backbone + asymmetric stretching vibrations of C-O-C bonds in the backbone
**1305**	Twisting vibrations of CH_2_ groups
**1441**	Scissoring vibrations of CH_2_ groups
**1724**	Stretching vibrations of C=O bonds
**2918**	Symmetric stretching vibrations of CH_2_ groups
**PLA**	
**402**	Wagging vibrations of C-O-C bonds in the backbone + rocking vibrations of O-C=O bonds
**874**	Symmetric stretching vibrations of C-O-C bonds in the backbone
**1454**	Scissoring vibrations of CH_3_ groups
**1768**	Stretching vibrations of C=O bonds
**2948**	Symmetric stretching vibrations of CH_3_ groups
**3002**	Asymmetric stretching vibrations of CH_3_ groups + stretching vibrations of C-H bonds

**Table 3 polymers-14-05367-t003:** Ratios of the peak intensities of PMeOEP bands (at 737 and 2963 cm^−1^) and PCL bands (at 1109, 1724 and 2918 cm^−1^) for the PCL—b—PMeOEP copolymers with various mole contents of PMeOEP (χ_PMeOEP_) and PCL (χ_PCL_). Raman spectra of the samples were recorded with the excitation wavelength of 532 nm.

χ_PMeOEP_: χ_PCL_	I_737_/I_1724_	I_737_/I_1109_	I_2963_/I_2918_	I_2963_/I_1724_	I_2963_/I_1109_
**0:1**	~0	~0	—	—	—
**0.05:0.95**	0.07	0.05	0.26	1.8	1.4
**0.06:0.94**	0.05	0.04	0.24	1.8	1.3
**0.15:0.85**	0.21	0.17	0.30	2.2	1.8
**0.16:0.84**	0.28	0.21	0.32	2.5	1.9
**0.23:0.77**	0.32	0.25	0.35	2.7	2.1
**0.86:0.14**	7.2	2.6	1.7	26	9.3
**1:0**	—	5.4	7.6	—	16

**Table 4 polymers-14-05367-t004:** Ratios of the peak intensities of PMeOEP bands (at 737 and 2963 cm^−1^) and PCL bands (at 1109, 1724 and 2918 cm^−1^) for the PCL—b—PMeOEP copolymers with various mole contents of PMeOEP (χ_PMeOEP_) and PCL (χ_PCL_). Raman spectra of the samples were recorded with the excitation wavelength of 785 nm.

χ_PMeOEP_: χ_PCL_	I_737_/I_1724_	I_737_/I_1109_	I_2963_/I_2918_	I_2963_/I_1724_	I_2963_/I_1109_
**0:1**	~0	~0	—	—	—
**0.05:0.95**	0.17	0.08	0.23	0.24	0.12
**0.06:0.94**	0.13	0.06	0.19	0.20	0.09
**0.15:0.85**	0.55	0.28	0.28	0.29	0.15
**0.16:0.84**	0.70	0.36	0.26	0.28	0.14
**0.23:0.77**	0.81	0.42	0.31	0.34	0.18
**0.86:0.14**	17	4.0	1.6	3.3	0.77
**1:0**	—	20	6.1	—	3.2

## Data Availability

Not applicable.

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
