# Peer review of "Raman Study of Block Copolymers of Methyl Ethylene Phosphate with Caprolactone and L-lactide"

_polymers, 2022, doi:10.3390/polym14245367_

Round 1
Reviewer 1 Report
In this manuscript, the authors investigated the Raman spectra of block copolymers of methyl ethylene phosphate (MeOEP) with caprolactone (CL) and L-lactide (LA), and analyzed possibilities of Raman spectroscopy for quantitative description of the structure of these copolymers.
1. In abstract, the expression of experimental findings and research results is not clear. The readers can not clearly see what the results are from the experiment. If possible, Please modify.
2. For PMeOEP-b-PLA copolymer, there is only one set of sample. If there are multiple sets of samples, it is more convincing. If possible, the author supplements the data of several different sets of samples.
3. The conclusion is not concise enough, which should be refined. In addition, the significance of this study and inspiration for subsequent research should be added.
4. Whether, the crystallinity of the block copolymers affects the quantitative description of the structure of the copolymer.
Reviewer 2 Report
The manuscript offers some good insights on vibrational spectroscopy of Polyester-methyl ethylene phosphate block copolymers and their synthesis. However, it lacks novelty and compelling content, which would make it more interesting and more likely to be cited. In addition to this, the manuscript needs extensive English language revision, especially in the introduction section.
With a much more detailed description of the quantum chemical calculations and the relative results, this paper would feel much more compelling and complete, whilst at the present state it feels more like a large appendix to the extensive computational chemistry work the authors must have carried out.
It's difficult to identify what compelling question this work addresses and what novelty defines it, on a well-established characterisation technique and known materials. Despite the scientific soundness, the details given on the synthesis and the vibrational spectroscopy analysis of the compounds and the consistency of conclusion and results, the reviewer disbelieves that this work would contribute significantly to the subject area. It's not original or relevant enough to warrant publication.
Round 2
Reviewer 1 Report
The author has made the necessary revise as suggested by the reviewers and the manuscript can be ready for publication.
Reviewer 2 Report
The comments and the amendments made by the authors denote good will and concrete effort in improving the paper.
In the manuscript you state that : "To assign the Raman bands of PLA, PCL and PMeOEP we carried out the quantum 187 chemical calculations of the optimized geometries and Raman spectra of oligomers of 188 these polymers, using as the models oligomers consisting of from 4 to 9 monomeric units 189 (for details, see Figures S11−S13 and Tables S1−S3 in the Supplementary Materials"
However, in the supplementary file I received there's still no trace of fig. S11 - 13 and Table S1 -3.
Please provide the updated supplementary file and the manuscript can then be published.